# Hydrodynamic Analysis of the Clinical Findings in Pachychoroid-Spectrum Diseases

**DOI:** 10.3390/jcm11175247

**Published:** 2022-09-05

**Authors:** Okihiro Nishi, Tsutomu Yasukawa

**Affiliations:** 1Jinshikai Medical Foundation, Nishi Eye Hospital, 4-14-26 Nakamichi, Higashinari-ku, Osaka 537-0025, Japan; 2Department of Ophthalmology and Visual Science, Nagoya City University Graduate School of Medical Sciences, Nagoya 464-0083, Japan

**Keywords:** central serous chorioretinopathy, pachyveins, pachychoroid, Bernoulli’s Principle

## Abstract

We wish to demonstrate that theorems of fluid dynamics may be employed to hydrodynamically analyze the clinical presentations seen within the pachychoroid-spectrum diseases (PSD). **Methods:** We employed both the Equation of Continuity Q = A · V in which Q represents blood flow volume, A the sectional area of a vessel, and V blood flow velocity as well as Bernoulli’s Principle 1/2 V^2^ + P/ρ = constant where V represents blood flow velocity, P static blood pressure and ρ blood density. The Equation of Continuity states that a decrease in flow volume occurs simultaneously with a decrease in the flow velocity and/or sectional area, and vice versa. Bernoulli’s Principle states that a decrease in the velocity of a fluid occurs simultaneously with an increase in static pressure, and vice versa. **Results:** Hyperpermeability of the choriocapillaris, as visualized on fluorescein angiography and indocyanine green angiography (ICGA), causes a fluid exudation and, therefore, a decrease in the blood flow volume Q which elicits a simultaneous decrease in the blood flow velocity V clinically observable in filling delay into the choriocapillaris on ICGA. An increase in the static blood pressure P will simultaneously occur in venules in accord with Bernoulli’s Principle. **Conclusions:** A decrease in the blood flow velocity in the choriocapillaris due to its hyperpermeability will hydrodynamically elicit an increase in the blood pressure in venules. This blood pressure rise may expand Sattler and Haller veins, forming pachyveins. The primary lesion of PSD can be in pigment epithelium and choriocapillaris.

## 1. Introduction

Pachychoroid spectrum diseases [1] (PSD) include a variety of clinical findings such as central serous chorioretinopathy (CSC) with serous retinal detachment (SRD) [2,3] of the sensory retina, choroidal vascular hyperpermeability [4], late filling of dye into the choroid on indocyanine green angiography (ICGA) in choroid [4], pachychoroid [3], pachyveins [5], thinning (attenuation) of the choriocapillaris layer, anastomosis in vortex veins [4,6], thickened sclera [7], loculation of fluid in suprachoroid [8,9], and pigment epitheliopathy [10], all of which can result in not only CSC but also pachychoroid neovasculopathy (PNV) [11] and polypoidal choroidal vasculopathy (PCV) [12], similar in appearance to age-related macular degeneration (AMD).

Some theories as to the pathogenesis of these entities have been proposed. Kishi et al. [4] found filling delay areas in the choriocapillaris and dilation of the emissary vortex vein on ICGA and suggested that the blood flow into the choriocapillaris is delayed as a result of congestion of the dominant vortex veins. Imanaga et al. [7] found sclera thickening in CSC and postulated that the scleral thickening may choke the emissary vortex veins, eliciting the emissary vortex vein congestion. However, this vortex vein congestion theory cannot consistently explain all the above-mentioned characteristic findings. Although this theory may suggest that pachyveins result from blood congestion caused by choking the vortex veins, it is still questionable whether longstanding congestion in choriocapillaris, venules, and vortex veins elicit vascular endothelial growth factor (VEGF)-production which may finally cause PNV and PCV. Dropout of choriocapillaris is unlikely to occur by compression of a pachyvein, because capillary pressure is higher than venous pressure. On the other hand, there will be another possibility that pigment epitheliopathy as a primary lesion may alter VEGF-production, followed by choroidal vascular hyperpermeability and incidence of CSC, PNV, and PCV. This hypothesis needs to explain that alteration of retinal pigment epithelium and choriocapillaris can elicit other characteristic findings such as pachychoroid and pachyveins.

Since most of these findings are related to the blood flow in the choroid, there is a possibility that a hydrodynamic analysis may provide clues as to the possible common pathogenesis to the entities we name PSD.

We attempted to analyze and interpret these findings based on the physics theorems of Equation of Continuity and Bernoulli’s Principle of fluid mechanics [13].

## 2. Methods

We applied the two following representative theorems of fluid dynamics in physics to the above clinical findings and phenomena.

Equation of Continuity [13]: Q = A · V

Q: (blood) flow volume, A: sectional area of vessel, V: (blood) flow velocity.

This equation is based on the principle of “Conservation of mass”. Thus, for a fluid through a pipe at all the cross-section, the quantity of fluid per second is constant. As the equation shows, a decrease in the flow volume occurs simultaneously with a decrease in the flow velocity and/or sectional area, and vice versa.

2.Bernoulli’s Principle [13]: 1/2 V^2^ + P/ρ = Constant

V: (blood) flow velocity, P: static (blood) pressure, ρ: (blood) fluid density

This equation relates velocity, static pressure, and elevation changes of a fluid in motion. The equation can be obtained by integrating Euler’s equation along the streamline for a constant fluid.
1/2 V^2^ + g · z + P/ρ = Constant

Since gravitational g and elevation z for the choroid are a small part of papillo-macular region and always same for a patient during the examination, they can be regarded as constant. Hence, the equation can be formed as
1/2 V^2^ + P/ρ = Constant

As the equation shows, a decrease in the velocity of a fluid occurs simultaneously with an increase in static pressure, and vice versa.

These theorems can be applied to non-viscous, noncompressible fluids [13]. The viscosity of a fluid can be expressed as the coefficient of viscosity [14]: water 1 cp (centi poise), blood plasma 1.2–1.3 cp, blood 3.5–5.0 cp, olive oil 80 cp [15]. Since blood is similar in its consistency to water, it can be regarded as non-viscous. Noncompressibility is defined as the ratio of the speed of a flow to the speed of sound and expressed in Mach number [13,16]. Accordingly, when the speed is less than Mach 0.3, the fluid is noncompressible [16]. Blood flow can be considered to meet this definition (arterial velocity: 1.9–19.0 cm/s, venous velocity: 1.5–7.1 cm/s in the index finger [17]).

## 3. Results

Applying the variety of clinical presentations of PSD to these physics equations, it is important to first recognize that the hyperpermeability [4] of the choriocapillaris will elicit blood plasma exudation into the interstitial space surrounding the choriocapillaris forming pachychoroid driven by oncotic pressure of exudates including albumin and fibrinogen. The concomitant increase of hydrostatic pressure in the interstitial space of the choroid, when accompanied by the disruption of blood retinal barrier in the retinal pigment epithelium, will lead to exudation and/or transudation into the sensory retina, forming SRD. This may result in a decrease in blood flow volume (Q) in the choriocapillaris, since the exudate -in contrast to the transudate- will not be totally absorbed into the choriocapillaris at the arteriole side. A decrease in (Q) simultaneously elicits a decrease in the blood flow velocity (V), according to the Equation of Continuity. This decrease in the blood flow velocity is clinically observable as filling delay into choriocapillaris on ICGA [4] and on laser speckle flowgraphy (LSFG) (Figure 1 and Figure 2).

When a decrease in the blood flow velocity (V) exists, Bernoulli’s Principle dictates that an increase in the static blood pressure (P) will simultaneously occur in the venules. Furthermore, since the blood density ρ may increase due to the exudation, its reciprocal, the blood pressure P will become much greater to keep the Bernoulli’s equation constant. With the decelerated velocity of the blood flow, the elicited higher blood pressure may expand Sattler and Haller vessels, forming pachyveins.

## 4. Discussion

Applying these two physics theorems of fluid mechanics to the large variety of clinical findings in the PSD entities, one must consider mathematical-physical equations and relate them to the hyperpermeability of the choriocapillaris and disruption of the blood retinal barrier of the pigment epithelium with CSC. The two theorems lead to the conclusion that pachyveins may be the result of a blood pressure increase occurring simultaneously in the wake of a decrease in the blood flow velocity in the choriocapillaris and choroidal venules. Therefore, hyperpermeability of the choriocapillaris including pigment epitheliopathy may be considered as the primary lesion of PSD [20,21,22]. Lipid accumulation in Bruch’s membrane initiates at 30 years of age, thereafter, decreasing the hydraulic conductivity of Bruch’s membrane with aging [20,21]. Schmidt-Erfurth et al. [22] showed that lipid accumulation in Bruch’s membrane enhanced expression of VEGF under the retinal pigment epithelium in low density lipoprotein receptor knockout mice, an experimental model of lipid accumulation in aging eyes. We speculate that lipid accumulation may be associated with upregulation of VEGF secreted from the retinal pigment epithelium and focal pigment epitheliopathy may affect amount and dynamics of VEGF accumulated between the retinal pigment epithelium and Bruch’s membrane, resulting in both attenuation and hyperpermeability of the choriocapillaris. In truth, CSC generally develops over 30 years of age. Moreover, retinal pigment alteration seems to be necessary for PSD, while either pachychoroid or pachyveins is not always present in all cases of PSD. Thus, the pachychoroid including pachyveins alone is not the primary lesion rather the secondary accompanying phenomenon in these disorders.

The effect of the vortex vein compression and its consequence by the thickened sclera [7,8,9] can be explained using the principles discussed here. The compression of the vortex vein delays the blood flow velocity (V). According to the Bernoulli’s Principle, the resulting delayed blood flow will elicit simultaneously a blood pressure (P) rise which can expand Sattler and Haller vessels. However, whether the compression of the vortex veins as the primary cause can elicit pigment epitheliopathy followed by VEGF-production, PNV, and PCV is unanswered. The thickened sclera and loculation of fluid in suprachoroid are explainable by diffusion of oncotic pressure-driven exudates such as albumin in the interstitial space of the choroid, because even molecules as large as antibodies can pass through the sclera [23].

All of above considerations emphasize the important role of the blood pressure in pachyveins and pachychoroid formation which, to date, has received scant attention. When, in the future, the means to ascertain the blood pressure in the Haller layer becomes available, the above theoretical considerations can be corroborated. We are now developing a method to measure the blood pressure of choroid using LSFG.

The arising important issue will be what then and where the primary lesion of PSD is. The choking of vortex veins eliciting congestion in the choroidal circulation [7] could be a prime consideration. The congestion by the thickened sclera alone may be able to initially raise the blood pressure in the choriocapillaris and Haller layer. When this raised pressure exceeds the so-called plasma-osmotic pressure in the choriocapillaris, the transudate around the choriocapillaris can no longer be absorbed into the capillaries, resulting in a condition of chronic edema. However, whether this condition can be the cause of pigment epitheliopathy, hyperpermeability of choriocapillaris and choroidal neovascularization, resulting in the exudation into the interstitial space is still unanswered.

On the other hand, even in eyes without the primary vortex vein congestion by the thickened sclera, when the blood plasma components exit the choriocapillaris due to its increased permeability, the blood flow slows down, and the blood pressure, simultaneously, rises hydrodynamically in accord with Bernoulli’s Principle resulting in an expansion of Haller veins, as shown in this study. Thus, the blood pressure rises and congestion in the choroidal circulation can occur as a secondary result of the hyperpermeability of the choriocapillaris and the disruption of the blood retinal barrier. The primary lesion then would be the pigment epithelium, Bruch’s membrane and/or choriocapillaris [18]. In this case, the thickened sclera would then be considered as secondary lesions. It could be caused by an imbibition of the edema and/or interstitial exudates into the corresponding sclera region.

Finally, the limitation of our study is that the above results and discussions are based merely on the theoretical considerations. In the future studies, we will measure the difference in the blood flow speed in the choroid of the macula region, as shown in the Figure 1 and Figure 2, using LSFG in a cohort of the patients with PSD. The blood pressure will be measured using LSFG as well. These may help corroborate the theory.

## 5. Conclusions

According to Bernoulli’s Principle of fluid mechanics, a decrease in the velocity of blood flow in the choriocapillaris in CSC and PSD occurs simultaneously in an increase in the blood pressure of the vortex veins, which may expand the venules. Pachyveins can be the result of a blood pressure increase occurring simultaneously in the wake of a decrease in the blood flow velocity in the choriocapillaris, suggesting that pigment epitheliopathy including hyperpermeability of the choriocapillaris may be the primary causative lesion of pachychoroid-spectrum diseases.

## Figures and Tables

**Figure 1 jcm-11-05247-f001:**
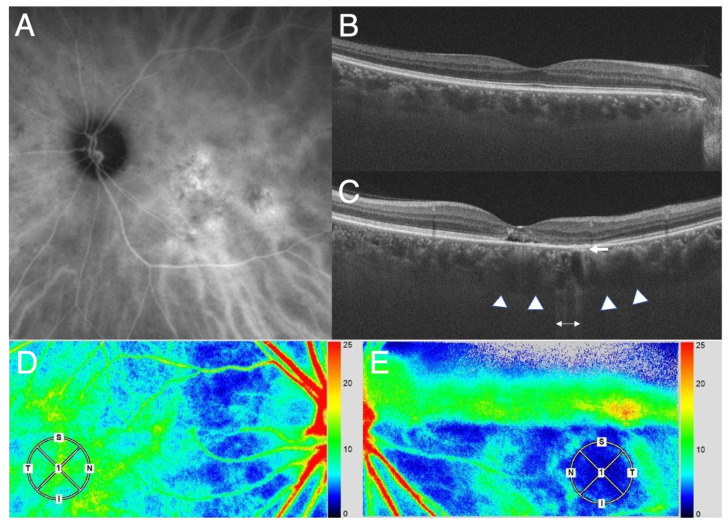
Photographs of the affected eye and the healthy fellow eye of a patient with CSC. (**A**). Indocyanine green angiography shows choroidal vascular hyperpermeability in the macula region. (**B**,**C**). Horizontal section on optical coherence tomography in the healthy right (**B**) and the affected left eye with CSC (**C**). The CSC eye shows pachyveins and pachychoroid (arrow heads) and attenuation of the choriocapillaris (arrow) at the area with pigment alteration suggested by enhanced light transmission (two-way arrow). (**D**). LSFG in the healthy right eye. The color bar (at the right-side margin) indicates the blood flow speed. The speed is faster toward red and slower toward blue. Note that the blood flow speed in the macula region (within the circle) reflects choroidal blood flow signal because of the lack of retinal vasculature and is only slightly lower compared to that at the adjacent area, which may indicate physiological condition. (**E**). LSFG in the affected left eye. Note that the blood flow speed in the area with choroidal vascular hyperpermeability in A involving the macula region (within the circle) is extremely slow (indicated by blue) compared to that in the surrounding area. Thus, there is a large decrease in the blood flow speed in the area affected by CSC, indicating that there will be a blood pressure rise in the region according to Bernoulli’s theorem.

**Figure 2 jcm-11-05247-f002:**
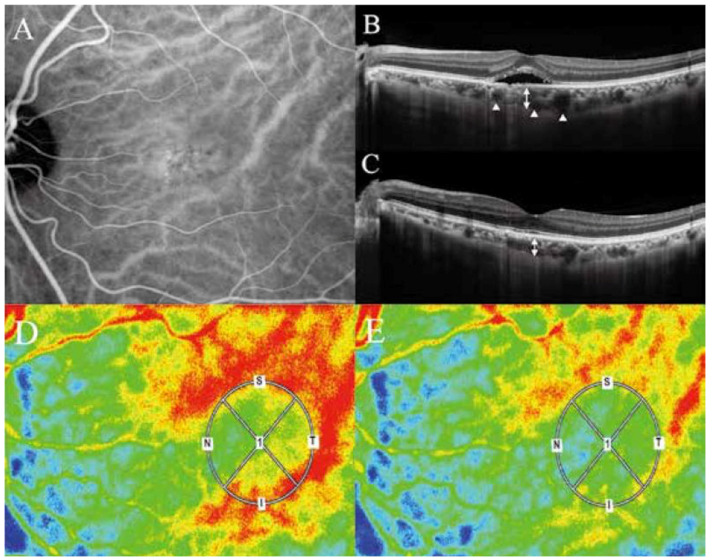
Photographs of an eye with CSC. (**A**). Indocyanine green dye leakage at the macula region shows hyperpermeability of the choriocapillaris. (**B**). CSC, pachychoroid, and pachyveins during the onset. (**C**). Regression of CSC in the same eye. (**D**). LSFG at the onset of CSC. The LSFG shows blood flow velocities in the choroid at the macula region where SRD exists (within the circle) and its surrounding area of the choroid. Note that the velocity in the macula region is much slower (yellow to green) than that in the surrounding area (mostly red), suggesting that there is a drastic decrease of the choroidal blood flow speed in the macula region. (**E**). The choroidal velocities after the regression of SRD. Note that there is a much less difference in the speed between the macula region and its surrounding area. Less difference in the speed means less difference in the blood pressure between the two regions, according to Bernoulli’s theorem, reducing pachyveins. (Adapted from Saito W et al., 2020 [18] and 2021 [19]).

## Data Availability

The study did not report any data.

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
