# Peer review of "Hydrodynamic Analysis of the Clinical Findings in Pachychoroid-Spectrum Diseases"

_jcm, 2022, doi:10.3390/jcm11175247_

Round 1

Reviewer 1 Report

The article attempts to apply physics theorems to the pachychoroid spectrum. However, the introduction does not convey the need for the study. Additionally the methods and results section fall short in explaining what has been done. It would have more translational value if the authors discussed some practical cases with longitudinal data how their theory translates into the actual changes in the eye. The discussion as well needs improvement. 

Author Response

Thank you so much for your valuable comments and suggestions to our manuscript. Accordingly, we have revised the manuscript and will explain the details of revisions and our responses to the reviewers’ comments, point by point.

1. We have added some statement to the Introduction to explain the necessity of the study.

2. We have provided two Figures in the Results to ease the comprehension of our manuscript.

3. At the end of Discussion, we added the statement for the limitation of the study.

Reviewer 2 Report

Dear author,

you presented an interesting communication to the scientific community. Paper is well written and brings a new idea in the phisiopathology of pachycoroid, based on physics.

It would be great if you could provide some art/animation to turn easier the comprehension of what you present.

Author Response

Thank you so much for your valuable comments and suggestions to our manuscript. Accordingly, we have revised the manuscript and will explain the details of revisions and our responses to the reviewers’ comments, point by point.

We have provided two Figures.

Reviewer 3 Report

the paper has been reviewed and I have some concerns:

the idea is interesting, but there is no concrete evidence of the authors' theory.

The methods are not clear. The study was done on pictures of how many eyes?

The conclusions are interesting, but the methods and results need to be clarified.

Author Response

Thank you so much for your valuable comments and suggestions to our manuscript. Accordingly, we have revised the manuscript and will explain the details of revisions and our responses to the reviewers’ comments, point by point.

To ease the comprehension, we have provided two Figures in the Results. On your comments that there is lack of explanations, we have added some additional sentences to Introduction, Methods, Results, and Discussion. They are underlined for your recognition.

Reviewer 4 Report

This communication is very interesting in its approach. It lacks however major explanations and does not justify many assumptions.

- viscosity depends on temperature

- equation of continuity must accept the assumptions that the fluid is homogenous, but blood has a laminar flux and a viscosity that depends also on temperature. The blood flow must be calculated integrating Poiseuille's equation. 

- equation formulas and diagram representations are lacking

Author Response

Thank you so much for your valuable comments and suggestions to our manuscript. Accordingly, we have revised the manuscript and will explain the details of revisions and our responses to the reviewers’ comments, point by point.

1. I agree with your comment that viscosity depends on temperature. However, the temperature of the choroid may be the same with the body temperature around 36.5ËšC, and it will be the same within the corresponding part of the choroid, so that the temperature may not affect the application of the theorems and the results. We thought that Poiseuille’s equation cannot be applied for the situation, because it is applied only for laminar flow with low Reynolds number. The blood flow in the choriocapillaris, venules, and Haller veins may have low value of Reynolds number, which may allow the vessels laminar flow. But, in the choroid, the vessels are not straight long enough, much branched and tangled, so that the blood flow there will not in laminar flow but rather in disturbed flow (turbulence). Therefore, we thought that it is not appropriate to apply Poiseuille’s equation to the choroidal blood flow.

2. On your comments, we have added some additional sentences to Introduction, Methods, Results and Discussions. They are underlined for your recognition.